

**Fungal loop transfer of N depends on biocrust constituents and N form**

**Zachary T. Aanderud[1], Trevor B. Smart[1], Nan Wu[2], Alexander S. Taylor[1],**

**Yuanming Zhang[2], Jayne Belnap[3]**
[1]Department of Plant and Wildlife Sciences, Brigham Young University, Provo, Utah

       84602, USA

[2]Xinjiang Institute of Ecology and Geography, Key Laboratory of Biogeography and

       Bioresource in Arid Land, Chinese Academy of Sciences, Urumqi 830011, China

[3]US Geological Survey, Southwest Biological Science Center, 2290 S. Resource

       Blvd., Moab, UT 84532, USA

**Running head:** dark septate fungi translocate ammonium in biocrusts

**Key words:** ammonium, Ascomycota, Colorado Plateau, dark septate endophyte,

fungal loop, Indian ricegrass, Pleosporales

Correspondence to: Zachary T. Aanderud (zachary_aanderud@byu.edu)



**Abstract.**      Besides performing multiple ecosystem services individually and

collectively, biocrust constituents may also create biological networks connecting

spatially and temporally distinct processes. In the fungal loop hypothesis, rainfall

variability allows fungi to act as conduits and reservoirs, translocating resources

between soils and host plants. To evaluate the extent biocrust species composition and

N form influence loops, we created a minor, localized rainfall event containing

$^{15}NH_4^+$ and $^{15}NO_3^-$ and measured the resulting $\delta^{15}N$ in surrounding cyanobacteria- and

lichen-dominated crusts and grass, *Achnatherum hymenoides*, after twenty-four hours.

We also estimated the biomass of fungal constituents using quantitative PCR and

characterized fungal communities by sequencing the 18S rRNA gene. We only found

evidence of fungal loops in cyanobacteria-dominated crusts where $^{15}N$, from $^{15}NH_4^+$,

moved 40 mm h$^{-1}$ and the $\delta^{15}N$ in crusts decreased as the radial distance from the

water addition increased (linear regression analysis: $R^2=0.58$, $F=16$, $P=0.002$, $n=14$).

In cyanobacteria crusts, $\delta^{15}N$, from $^{15}NH_4^+$, was diluted as Ascomycota biomass

increased (linear regression analysis: $R^2=0.50$, $F=8.8$, $P=0.02$, $n=14$), Ascomycota

accounted for 82% (±2.8) of all fungal sequences, and one order, Pleosporales,

comprised 66% (±6.9) of Ascomycota. The lack of loops in moss-dominated crusts

and substantial movement of $^{15}NO_3^-$ may stem from mosses effectively sequestering

newly fixed N and fungi preferring $^{15}NH_4^+$ for amino acid transformation and

translocation. No label entered *A. hymenoides*. Our findings suggest that minor

rainfall events allow dark septate Pleosporales to rapidly translocate N in the absence

of a plant sink.





## 1 Introduction

Fungi may act as conduits for biological networks connecting belowground ecosystem processes to plants. Soil fungi contribute to all aspects of litter decomposition through the generation of a myriad of extracellular enzymes (Osono 2007, Schneider et al. 2012); altered trophic dynamics, decomposer species diversity, and nutrient turnover rates (Hattenschwiler et al. 2005); and by forming multiple types of endophyte-plant symbioses (Johnson et al. 1997, Saikkonen et al. 2004). Endophytic fungi ,in particular, form hubs connecting spatially and temporally distinct microbial-mediated soil processes and plants. For example, the pervasive distribution of mycorrhizae in mesic systems allows common mycorrhizal networks to deliver essential resources, which promotes or hinders seedling growth depending on the network species composition (van der Heijden and Horton 2009), and facilitates the one-way transfer of multiple forms of N and P between two plant species linked by arbuscular mycorrhizae and ectomycorrhizae (He et al. 2003, Walder et al. 2012). In xeric systems, endophytic fungi are also implicated in moving resources within biological networks in a theory known as the fungal loop hypothesis. The hypothesis states that fungi, supported by biotrophic C from plants and cyanobacteria, act as intermediate reservoirs transforming and translocating resources between soils and plants (Collins et al. 2008, 2014) ). Perhaps the most notable example of a fungal loop, albeit from a limited number of studies, occurred in fungal-dominated cyanobacteria biocrusts from a Chihuahuan Desert grassland. Specifically, $^{15}NO_3^-$ applied to a root-free biocrust rapidly moved into the perennial





64 grass, *Bouteloua* species, up to 1 m away within 24–hours (Green et al. 2008).

Furthermore, $^{13}$C-labeled glutamic acid applied to leaf surfaces of *Bouteloua* was

66 found in biocrusts. Despite the intriguing evidence, many aspects of this burgeoning

hypothesis remain to be validated (Collins et al. 2014).

68

Biocrust composition and soil moisture availability interactions may dictate the

70 movement of resources in fungal loops. Desert fungal-plant interactions occur across

spatially discontinuous patches of vegetation interspersed by patches of soils

72 colonized by biocrusts (Belnap et al. 2005). Fungi participating in loops are

necessarily associated with a mosaic of other biocrust organisms (i.e., cyanobacteria,

74 green algae, lichens, mosses, and other bacteria). The metabolic activity of biocrust

constituents participating in fungal loops, including plants, are moisture-dependent

76 and regulated by the magnitude and seasonality of episodic rainfall events. A

pulse-reserve paradigm (Collins et al. 2008) may explain biological activities where

78 minor rainfall pulses stimulate microorganisms, generating reserves of resources to be

exploited during subsequent rainfall events (Huxman et al. 2004, Welter et al. 2005).

80 In such loops, minor rainfall events may stimulate $N_2$ fixation by free or

lichen-associated cyanobacteria (Belnap et al. 2003), N mineralization by bacteria and

82 fungi (Cable and Huxman 2004, Yahdjian and Sala 2010) and nitrification and possibly

denitrification (Wang et al. 2014) all increasing the levels of $NH_4^+$ or $NO_3^-$. Fungal

84 species, including fungal endophytes, may compete with mosses, lichen,

cyanobacteria, and other bacteria for newly released N. Once sequestered, the N may be



transformed into amino acids and transported within mycelium (Jin et al. 2012, Behie

and Bidochka 2014). Larger rainfall events may then activate plants, allowing the host

to receive N from the fungi and transfer photosynthate to the fungal endophyte. If

fungal endophytes are poor competitors for newly released N, preferentially sequester

one inorganic N form over another, or more efficiently transform and transport $NH_4^+$ or

$NO_3^-$, biocrust constituents and N form may influence the translocation of N in fungal

loops.
The fungal endophytes most likely involved in the loop hypothesis are dark septate

fungi. Few arbuscular mycorrhizal fungi are found in biocrusts (Porras-Alfaro et al.

2011) or as endophytes in desert plants (Titus et al. 2002), due to mycorrhizae being

relatively sensitive to dry soil conditions (Aguilera et al. 2016). In contrast, the

majority of biocrust fungi are Ascomycota, with the Pleosporales being widespread

and dominant (Bates et al. 2012, Porras-Alfaro et al. 2011). Pleosporales, along with

other Ascomycota fungal orders, contain dark septate endophytes (Jumpponen and

Trappe 1998). Dark septate are thermal- and drought-tolerant fungi due to

melanin-rich cell walls conferring protection from UV and drought stress (Gostincar

et al. 2010). Taken together, the prevalence of dark septate fungi in desert systems,

along with their ability to maintain metabolic activity under low water potentials

(Barrow 2003), makes these endophytes excellent candidates to translocate resources

in loops (Green et al. 2008).





Minor rainfall events may allow fungi to act as conduits and reservoirs for N. To

investigate the potential for biocrust constituents and N form to influence the

movement of N through the putative fungal loops, we created minor, localized rainfall

events and measured $\delta^{15}N$, from $^{15}N\text{-}NH_4^+$ and $^{15}N\text{-}NO_3^-$, within surrounding

cyanobacteria- and moss-dominated crusts, and grass, *Achnatherum hymenoides*

(Indian ricegrass). In tandem with $^{15}N$ analyses, we estimated the biomass of two

major division of fungi (Ascomycota and Basidiomycota) and bacteria, and

characterized fungal communities by sequencing the 18S rRNA gene to identify

potential link between fungal taxa and $^{15}N$ movement.
**2    Materials and methods**

        **2.1 Site description**

We conducted our study in two cold desert ecosystems of the Colorado Plateau, UT.

One site was near Castle Valley (40°05'27.43"N-112°18'18.24"W) and the other was

adjacent to the US Geological Survey (USGS), Southwest Biological Science Center

Research Station in Moab, UT (40°05'27.43"N-112°18'18.24"W). Rugose crusts

consisting of moss *Syntrichia caninervis* and cyano-lichens *Collema tenax* and

*Collema coccophorum* cover the Castle Valley site (Darrouzet-Nardi et al. 2015),

while smooth, light algal crusts of one cyanobacterium, *Microcoleus vaginatus*, cover

the USGS site. Specifically, biocrust cover across the Castle Valley was 50%

cyanobacteria, 22% *S. caninervis*, and 5-7% *Collema* spp. (Zelikova et al. 2012), and

100% cyanobacteria for the USGS. Across both sites, vegetation is dominated by



perennial grass *Achnatherum hymenoides* (Roem & Schult) and native perennial

shrub *Atriplex confertifolia* (Torr. & Frém). Mean annual temperature and

precipitation at Castle Valley is 13ºC and 269 mm, while the USGS site is slightly

warmer (MAT=13.8ºC) and drier (MAP=189 mm; based on 1981-2010 data; WRCC

2017). Both soils are Aridisols with Castle Valley classified as a sandy loam,

calcareous Rizno series (Darrouzet-Nardi et al. 2015) and USGS as a Bluechief series

sandy loam.
**2.2 Simulated rainfall events and $^{15}$N form applications**

We simulated rainfall events containing two isotopically-labeled, inorganic N forms

and tracked the movement of the label through our moss-dominated (Castle Valley)

and cyanobacteria-dominated biocrusts (USGS Station), and *A. hymenoides*. First, we

randomly selected six circular plots per site with a radius of 1.0 m and at least 10 m

apart from each other. Three plots were assigned to be labeled with K$^{15}$NO$_3$ (99 at.%)

and the other three plots to be labeled with ($^{15}$NH$_4$)$_2$SO$_4$ (99 at.%). Second, we

randomly selected five biocrust and five *A. hymenoides* along eight axes (e.g., N, NE,

E, SE, S, SW, W, and NW) radiating from the center of each circular plot and

measured the radial distance to biocrusts or grasses. Third, we simulated a 2.5 mm

rainfall event by spraying 3 mL of deionized water solution and either isotopic label

(0.30 mg $^{15}$N) onto a 5 cm diameter circle in the center of the circular plots (2 biocrust

types × 3 circular plots locations × 2 N forms × ≈10 samples [5 biocrusts or 4–8 *A.*

*hymenoides* depending on grass density in the circular plot]=137). The $^{15}$N additions

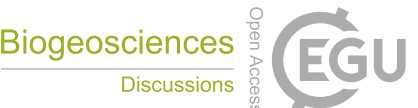



wetted the sandy loams (bulk density≈1.5 g cm$^{-3}$) to a depth of 1 cm and added

approximately equal $NH_4^+$ and double $NO_3^-$ concentrations to surface soils (Sperry et

al. 2006). All additions were completed midday in April as *A. hymenoides* were

starting to set seed.


**2.3 Sample collection and $^{15}$N analyses**

Biocrust and foliage samples were collected twenty-four hours after the simulated

rainfall event containing our different inorganic $^{15}$N forms. Biocrusts were removed

as three subsamples from each biocrust location with a soil corer (2 cm diameter × 5

cm length) to a depth of 2 mm. Crust distances away from the tracer application

ranged from 22–97 cm. The composited soil sample was kept cold (5°C) in the field,

split in the lab, and a portion of the soil was frozen (-20°C) until we performed fungal

and bacterial DNA analyses. We randomly selected five leaves from *Achnatherum*,

which ranged in distance anywhere from 29–120 cm away from the tracer application

and in volume from 0.002–0.048 m$^3$. The leaves and remaining soils (sieved 2 mm)

were air-dried, ground in a reciprocating tissue homogenizer, and analyzed for $^{15}$N

using a PDZ Europa ANCA-GSL elemental analyzer, interfaced with a PDZ Europa

20–20 isotope ration mass spectrometer (Sercon Ltd., Cheshire, UK) at the University

of California Davis Stable Isotope Facility (http://stableisotopefacility.ucdavis.edu).

We expressed the resulting isotope ratios in δ notation as parts per thousand (‰)

where:

$$\delta^{15}N = (R_{sample} / R_{standard}) \times 1000 \qquad\qquad (1)$$




$R$ is the molar ratio of the heavier to the lighter isotope ($^{15}N/^{14}N$) for the standard or

sample. To track the movement of inorganic N forms through our two biocrust types

(moss-lichen-dominated and cyanobacteria dominated biocrust) and into grasses, we

analyzed the relationships between $\delta^{15}N$ present in crust and leaf tissue to the distance

of the crust and *Achnatherum* by site using linear regression in SigmaPlot Version

13.0 (Systat Software, San Jose, CA).


**2.4 Biomass estimations of major fungal components**

To investigate the potential for fungi to translocate our $^{15}N$ forms, we estimated the

biomass of two major divisions of fungi (Ascomycota and Basidiomycota) and

bacteria in biocrusts using quantitative PCR. From the frozen biocrust samples, we

extracted genomic DNA using a DNeasy PowerLyzer PowerSoil Kit (Qiagen, MD,

USA) and quantified the gene copy numbers of Ascomycota and Basidiomycota on a

Mastercycler EP Realplex qPCR (Eppendorf, Hamburg, Germany) with SYBR Green.

We amplified division-specific regions of the internal transcribed spacer (ITS) with

primer pair ITS5 (forward) and ITS4A (reverse) for Ascomycota (Larena et al. 1999)

and ITS4B (forward) and 5.8sr (reverse) for Basidiomycota (Fierer et al. 2005). We

selected the universal bacterial 16S rRNA primer set EUB 338, forward, and Eub518,

reverse, to estimate the biomass of bacteria (Aanderud et al. 2013). In 12.5 µl

reactions, using KAPA2G Robust PCR Kits (KAPA Biosystems, Wilmington, MA,

USA), we amplified targeted genes using the following thermocycler condition: an

initial denaturation step at 94ºC for 3 min followed by 35 cycles of denaturation at



94ºC for 45 s, annealing at either at 55 ºC (Ascomycota), 64ºC (Basidiomycota), or

60ºC (bacteria) for 30 s, and extension at 72ºC for 90 sec. We generated qPCR

standards for Basidiomycota, Ascomycota, and bacteria from biocrusts using the

TOPO TA Cloning® Kit (ThermoFisher Scientific, MA, USA) as outlined by

Aanderud et al. (2013). The coefficients of determination ($R^2$) for our assays ranged

from 0.90 to0.99, and amplification efficiencies fell between 0.99 and 1.92. We

analyzed the relationships between biocrust $\delta^{15}N$ and the gene copy number of

Ascomycota and Basidiomycota using linear regression to investigate the potential for

fungi to act as conduits for inorganic N. We tested for differences in our biomass

estimates between the crust types using multiple t-tests and a Benjamini-Hochberg

correction to control for the false discovery rate associated with multiple comparisons

(Benjamini and Hochberg 1995).


**2.5 Biocrust fungal communities**

To identify the fungal taxa participating in N translocation, we characterized fungal

communities in biocrusts using bar-coded sequencing. We PCR amplified the V9

region of the 18S rRNA gene using a universal eukaryote primer set, 1391F and

EukBr, with a unique 12-bp Golay barcode fused to EukBr (Amaral-Zettler et al. 2009,

Hamady et al. 2008). Thermocycler parameters were similar to qPCR analyses and

consisted of a denaturation step at 94°C for 3 min, followed by 35 cycles of

denaturation at 94°C for 45 s, an annealing step at 57°C for 60 s, elongation at 72°C for

90 s, and a final extension at 72°C for 10 min. We then purified and pooled PCR





218     amplicon libraries to near equimolar concentrations using SequalPrep™ Normalization

        Plate Kits (Invitrogen, Carlsbad, CA, USA) and quantified the amplicon libraries by

220     real-time qPCR using a KAPA Library Quantification Kit (Kapa Biosystems,

        Wilmington, MA, USA). All samples were sequenced at the Brigham Young

222     University DNA Sequencing Center (http://dnasc.byu.edu/) using the Illumina HiSeq

        2500 platform (Illumina Biotechnology, San Diego, CA, USA), generating $2 \times 250$

paired-end reads. Illumina sequence reads were analyzed within QIIME (v. 1.9.1), an

        open-source software pipeline suitable for microbial community analysis (Caporaso et

al. 2010). We removed barcodes and primers with a custom, in-house script previous to

        joining paired-end reads by using fastq-join under default parameters (Aronesty 2011).

Joined reads were then de-multiplexed and checked for chimeras (Haas et al. 2011). We

        then clustered the de-multiplexed reads into OTUs, applying a similarity threshold of

97%, using QIIME's default OTU clustering tool-uclust (Edgar et al. 2011).

        Taxonomies of representative OTUs were assigned using uclust and the 18S rRNA

gene SILVA 128 database which was clustered into OTUs at 97% similarity (Quast et

        al. 2013). To evaluate if biocrust type supported similar fungal composition, we

calculated the relative recovery of 27 fungal orders, including dark septate lineages. We

        tested for differences between biocrust types using t-tests and a Benjamini-Hochberg

correction.
**3   Results**

        **3.1 Translocation of $^{15}NH_4^+$ in cyanobacteria biocrusts**





The movement of $^{15}$N was most dramatic in cyanobacteria-dominated biocrusts

following the addition of $^{15}$NH$_4^+$. In cyanobacteria crusts, $\delta^{15}$N decreased as the radial

distance from the central application point of $^{15}$NH$_4^+$ increased ($R^2$=0.58, $F$=16,

$P$=0.002, $n$=14, figure 1A). Surrounding the tracer application, $\delta^{15}$N was enriched

upwards of 40‰ more than 20 cm away and continued to be enriched to

approximately 10‰ almost 100 cm away. To a lesser extent, $^{15}$NO$_3^-$ followed a

similar pattern. $\delta^{15}$N decreased as the radial distance from the central application

point of $^{15}$NO$_3^-$ in cyanobacteria crusts increased, but the $\delta^{15}$N was never more

enriched than 8‰ ($R^2$=0.17, $F$=2.6, $P$<0.0001, $n$=15, figure 1B).

In moss-dominated biocrusts, there was no relationship between $\delta^{15}$N and the

radial distance from either the $^{15}$NH$_4^+$ ($R^2$=0.01, $F$=0.13, $P$=0.73, $n$=15, figure 1A) or

$^{15}$NO$_3^-$ addition ($R^2$=0.03, $F$=0.46, $P$=0.51, $n$=15, figure 1B). There was no

relationship between $\delta^{15}$N found in *A. hymenoides* leaves and the radial distance from

the $^{15}$NH$_4^+$ or $^{15}$NO$_3^-$ application with $\delta^{15}$N in leaves ranging from 3–18‰. The $R^2$

and $F$ values from the regressions between leaves $\delta^{15}$N and isotopic distance was

0.01–0.21 and 0.14–2.6 ($n$=14–23) respectively (data not shown).


**3.2 $^{15}$NH$_4^+$ movement in cyanobacteria biocrusts related to Ascomycota**

The biocrust that translocated N also exhibited a robust relationship between

Ascomycota biomass and biocrust $\delta^{15}$N. In cyanobacteria biocrusts, the greater the

gene copy number of Ascomycota the lower the $\delta^{15}$N from $^{15}$NH$_4^+$ ($R^2$=0.50, $F$=8.8,

$P$=0.02, $n$=14, figure 2A). Ascomycota biomass was marginally related to $\delta^{15}$N from



$^{15}NO_3^-$ in cyanobacteria crusts ($R^2$=0.07, $F$=1.1, $P$=0.08, $n$=15, figure 2B). In moss

crusts, however, there was no such relationship between $\delta^{15}N$ from $^{15}NH_4^+$ ($R^2$=0.05,

$F$=1.6, $P$=0.24, $n$=15, figure 2A) and $^{15}NO_3^-$ ($R^2$=0.01, $F$=0.10, $P$=0.75, $n$=15, figure

2B) and Ascomycota biomass. Basidiomycota and bacteria biomass in both crust

types was not related to either N form with $R^2$, $F$, and $P$ values ranging from 0.01–

0.10, 0.1–1.5, and 0.24–0.90 ($n$=12–15) respectively (data not shown). The biomass

of all measured biocrust components was consistently higher in moss- than

cyanobacteria-dominated crusts. Basidiomycota biomass was $1.5\times10^9$ ±$5.5\times10^8$

(means ±SEM) in cyanobacteria and $5.8\times10^9$ ±$7.2\times10^8$ in moss biocrusts (t-test, t=4.5,

$P$<0.0001, $df$=1, data not shown). Ascomycota biomass was $2.6\times10^7$ ±$4.5\times10^6$ (means

± SEM) in cyanobacteria and $1.1\times10^8$ ±$2.4\times10^7$ in moss biocrusts (t-test, t=3.3,

$P$=0.003, $df$=1). Bacterial biomass was at least two orders of magnitude lower in

biocrusts (cyanobacteria=$5.5\times10^6$ ±$8.9\times10^5$ and moss crusts=$2.7\times10^7$ ±$4.8\times10^6$,

(t-test, t =4.5, $P$<0.0001, $df$=1).


**3.3 Dark septate fungi as major components of biocrusts**

Four of the nine fungal orders contained known dark septate endophyte members and

were present in both biocrust types with the Pleosporales and Pezizales being dominant

taxa. In biocrusts: fungi comprised much of eukaryotic community

(cyanobacteria=30% ±4.7 and moss=33% ±4.0), Ascomycota was the most common

fungal division (cyanobacteria=82% ±2.8 and moss=87% ±2.9), and orders with

known dark septate members accounted for at least 67% of the Ascomycota



(cyanobacteria=83% ±4.8 and moss=67% ±8.6, figure 3). In cyanobacteria biocrusts,

Pleosporales accounted for 66% (±6.9) of all dark septates and the recovery of this taxa

was two-times higher in cyanobacteria- than moss- dominated crusts (t-test, t=03.0,

$P$=0.01, $df$=1). Even though the relative abundance of Pleosporales differed, the

number of gene copies of Pleosporales were similar between the two biocrusts

(cyanobacteria=$1.7 \times 10^7$ ±$6.3 \times 10^6$ and moss=$2.9 \times 10^7$ ±$1.3 \times 10^7$, t-test, t=0.99, $P$=0.35,

$df$=1) as determined by an extrapolation of qPCR values in conjunction with percent

recovery of taxa for Ascomycota. The *Pezizales* comprised a relatively larger

percentage of the biocrust community in moss-dominated biocrust with a recovery of

15% (± 8.3) and 28% (± 9.0) in cyanobacteria- and moss-dominated crusts respectively

(t-test, t=1.1, $P$=0.32, $df$=1). Eukaryotic community data was based on the recovery of

1,232,312 quality sequences and 5,176 unique OTUs.


### 4    Discussion

In biological networks, the magnitude and direction of resource transfer in fungi is

predominantly thought to be influenced by the physiological source-sink gradients

created by individual plants (Fellbaum et al. 2014) or between plants (Weremijewicz

et al. 2016). However, fungi may be more than just passive conduits and exert control

over resources due to their own sink-source resource needs (Simard and Durall 2004).

Our finding suggest that a minor rainfall event stimulated fungi, likely dark septate

endophytes, to rapidly translocate N at a rate of 40 mm h$^{-1}$ in the absence of a plant

sink for N. In the absence of a large rainfall event to stimulate plant activity, none of



the isotope entered *A. hymenoides*. The movement of N was only apparent in the

cyanobacteria-dominated crusts where $\delta^{15}N$ decreased as the distance from the

simulated rainfall event and $^{15}NH_4^+$ application increased. Further, the presence of

Ascomycota was related to biocrust $\delta^{15}N$ from $^{15}NH_4^+$ with the isotope being diluted

as Ascomycota biomass increased. Eighty-three percent of the Ascomycota were from

four fungal orders containing known dark septate endophytes and 66% of these taxa

were from one order, the Pleosporales. Taken together, our results suggest that fungal

loops are structured by fungal constituents, especially Pleosporales, translocating N

from $NH_4^+$ over $NO_3^-$.
**4.1 Fungal loops only in cyanobacteria-dominated crusts**

Although the moss, *S. caninervis*, appeared to hinder N transfer between biocrusts and

plants, our findings suggest that fungal loops do occur in cyanobacteria-dominated

biocrusts. Lichen-dominated biocrusts remain to be tested. Our results are consistent

with Green et al. (2008) whose previous work identified loops in cyanobacteria

biocrusts across the Chihuahuan Desert grassland and showed comparable distances

of N movement within biocrusts (Green et al. 2008=44 mm $h^{-1}$). Biocrust components

are known to fix and secrete up to 50% of their newly fixed C and 88% newly fixed N

to surrounding soils within minutes to days of fixation, depending on precipitation

characteristics (Belnap et al. 2003), and thus, would likely be available to other

biocrust constituents, such as fungi, for translocation. Bacteria and fungi were found

in both crust types, albeit in different amounts and species compositions, but mosses



mosses and lichens only occurred in one crust type. Mosses, in particular, change the

N cycling characteristics of arid lands. When *S. caninervis* was lost from this system,

a dramatic increase in $NH_4^+$, which ultimately nitrifies to $NO_3^-$, was observed (Reed et

al. 2012). The decomposition of dead mosses most likely contributed to the increase

of N; however, after the mosses died, inorganic N pooled in the remaining

cyanobacteria-dominated biocrust. Thus, mosses may be effective scavengers for N

and outcompete fungal endophytes for newly fixed N. The ability of mosses to

scavenge N is well recognized in other systems (Liu et al. 2013, Fritz et al. 2014).

Further, rhizoids, stem cells, and thalli of bryophytes may contain fungal associations

(Pressel et al. 2010). If desert mosses have fungal associations, then fungi have the

potential to move sequestered N to the mosses in a new kind of loop. Unlike plants

that may require a larger rainfall event to become active, fungi and mosses, including

*S. caninervis*, are stimulated by minor rainfall events (Wu et al. 2014) and dark

septate endophytes do colonize mosses (Day and Currah 2011). Thus, the exchange of

photosynthate and N may occur in a tighter, more localized loop. Another explanation

may lie in the microtopography of the two biocrusts. The moss-dominated crust was

pinnacled, while the cyanobacteria-dominated crust was smooth. Therefore, transport

distance between our application point and target plant was significantly further in

mosses than cyanobacteria crusts, potentially slowing the movement of N.
**4.2 Loops may preferentially move $NH_4^+$ over $NO_3^-$**



We found that $NH_4^+$, but not $NO_3^-$, was rapidly translocated within crusts. The

enrichment of $\delta^{15}N$, from $^{15}NH_4^+$, in cyanobacteria biocrusts was related to the

Ascomycota and potentially dark septate fungi due to their dominance. We explain

the negative relationship between Ascomycota gene copy number and $\delta^{15}N$ signal as a

simple dilution—the higher the biomass of Ascomycota, the more spread in the $^{15}N$

signal. Although the physiology of dark septate fungi remains relatively unexplored,

if our desert fungi are like other fungi then the preferential movement of $NH_4^+$ is

understandable. Generally, fungi prefer $NH_4^+$ over $NO_3^-$ (Eltrop and Marschner 1996),

as $NH_4^+$ is readily acquired by fungi and assimilated into amino acids. After $NH_4^+$

uptake and assimilation via the glutamate synthase or GS/GOGAT cycle, N is

incorporated into arginine through the urea cycle (Jin et al. 2012) due to the direct

assimilation of $NH_4^+$ into the GS/GOGAT pathway (Courtly et al., 2015). Thus, $NH_4^+$

is most likely transformed into arginine and moved within mycelium by amino acid

transporters (Govindarajulu et al. 2005, Garcia et al. 2016). Quantum dots

(fluorescent nanoscale semiconductors) have tracked the flow of organically derived

N into arbuscular mycorrhizae and into *Poa annua* in less than 24 hours (Whiteside et

al. 2009) and arbuscular colonization can also increase uptake of multiple other amino

acids (e.g., phenylalanine, lysine, asparagine, arginine, histidine, methionine,

tryptophan, and cysteine) by their host plants (Whiteside et al. 2012). $NO_3^-$ did move

in our cyanobacteria crusts but not nearly to the extent reported by Green et al.

(2008). Besides fungal preferences, other factors may play a role in the uptake of N,

such as the increase in mobility of $NO_3^-$ in soils, differences in soil cation exchange



capacity due to clay content, or fungi capitalizing on the more abundant N form

specific to a soil. More information is needed to identify the importance of N form

and the movement of organic N within fungal loops.


### 4.3 Dark septate and Pleosporales as conduits

Our results support the idea that Pleosporale*s* are the most likely conduits for N. Four

of the nine fungal orders we identified contained known dark septate endophyte

members but one order was the most abundant. The Pleosporales accounted for 66% of

the Ascomycota taxa in cyanobacteria crusts. Based on the relationship between $\delta^{15}N$

and Ascomycota biomass, the overwhelming abundance of Pleosporales, and the

universal occurrence of Ascomycota in biocrusts, the Pleosporales assumedly play a

role in fungal loops. We are not the first to reach this conclusion. Green et al. (2008)

also identified Pleosporale*s* as being the primary candidate involved in fungal loops. In

their semi-arid grassland, Pleosporale*s* were the most common taxa on *Bouteloua* roots,

in the rhizosphere, and in biocrusts. We found 799 operational taxonomic units, based

on 97% similarity, with all of the identifiable sequences, belonging to three genera:

*Leptosphaeria* (1.6% of Pleosporales sequences), *Morosphaeria* (3.8% of Pleosporales

sequences), and *Ophiosphaerella* (8.1% of Pleosporales sequences; data not shown).

*Leptosphaeria* and *Ophiosphaerella* may be pathogenic endophytes on grass species

(Martin et al. 2001, Yuan et al. 2017), but may also be beneficial by delaying and

reducing the symptoms of other fungal pathogens (Yuan et al. 2017). However, 86% of

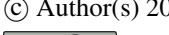



392 our Pleosporales taxa were unidentifiable and potentially novel, suggesting that much

remains unknown about dark septates in deserts.


**5. Conclusion**

396 Biocrusts, potentially, are interconnected in extensive biological networks. Dark

septate endophytes may act as conduits within the network by acting as both a sink

398 and source for translocating resources. In light of the absence of N movement in

moss-dominated crusts, mosses potentially hindered fungal loops. No isotopic label

400 entered *A. hymenoides* consistent with the fungal loop hypothesis that predicts plant

activity only after a larger rainfall event. Our results add to the indirect evidence of

402 fungal loops, but more information is needed to quantify the direct translocation of N

through dark septate fungi, characterize the magnitude and directionality of resources

404 within the endophytic relationship, and demonstrate the importance of a second larger

rainfall in structuring resource exchange.

406

**Acknowledgements** The portion of the research conducted by Dr. Wu and Dr.

408 Yuanming was funded by National Natural Science Foundation of China (grant #

41571256 and 41771299). Dr. Belnap thanks the Ecosystems Program of U.S.

410 Geological Survey. Any use of trade, firm, or product names is for descriptive

purposes only and does not imply endorsement by the U.S. Government.






**Author contributions** ZTA and JB designed the study. ZTA, TBS, NW, AST, and JB

conducted the experiments. ZTA, TBS, NW, AST, YZ, and JB analyzed and

interpreted the data. ZTA, TBS, NW, AST, YZ, and JB helped write and review the

manuscript. ZTA agrees to be accountable for all aspects of the work in ensuring that

questions related to the accuracy or integrity of any part of the work are appropriately

investigated and resolved.
**Conflict of interest** The authors declare no conflict of interest.





**Figure legend**

**Figure 1    Cyanobacteria biocrusts facilitated the translocation of N in**

**twenty-four hours.** Based on linear regression analyses, $\delta^{15}N$, from $^{15}NH_4^+$,

($R^2$=0.58, $F$=16, $P$=0.002, $n$=14) and to a lesser extent $^{15}NO_3^-$ ($R^2$=0.17, $F$=2.6,

$P$<0.0001, $n$=15), decreased as the radial distance from the isotopic application

increased. Values are $\delta^{15}N$ (‰) from two biocrust types, cyanobacteria- and

moss-dominated crusts, across circular plots (radius of 1.0 m) with a central

application (5 cm diameter circle) of $^{15}NH_4^+$ and $^{15}NO_3^-$.


**Figure 2    Ascomycota biomass influenced the distance N traveled.** In

cyanobacteria crusts, $\delta^{15}N$, from $^{15}NH_4^+$, was diluted as Ascomycota gene copy

number increased ($R^2$=0.50, $F$=8.8, $P$=0.02, $n$=14). Values are $\delta^{15}N$ (‰) from

biocrusts and Ascomycota gene copy numbers, an approximation of biomass, from

qPCR of the ITS region with primer pair ITS5 and ITS4A.


**Figure 3    Pleosporales were the dominant Ascomycota order and contained**

**dark septate species.** Pie chart values are means ($n$=6) of the relative recovery from

nine fungal orders, four of which contain dark septate endophytic taxa. Recovery was

based on OTUs from eukaryotic community libraries of the 18S rRNA gene (97%

similarity cutoff).




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

barcoded primers for pyrosequencing hundreds of samples in multiplex. Nature

Methods 5:235-237.

Hattenschwiler S, Tiunov AV, Scheu S. 2005. Biodiversity and litter decomposition

in terrestrial ecosystems. Annual Review of Ecology 36:191-218.

He XH, Critchley C, Bledsoe C. 2003. Nitrogen transfer within and between plants

through common mycorrhizal networks (CMNs). Critical Reviews in Plant

Sciences 22:531-567.

Huxman TE, Snyder KA, Tissue D, Leffler AJ, Ogle K, Pockman WT, Sandquist DR,

Potts DL, Schwinning S. 2004. Precipitation pulses and carbon fluxes in semiarid

and arid ecosystems. Oecologia 141:254-268.



Jin HR, Liu J, Huang XW. 2012. Forms of nitrogen uptake, translocation, and transfer

via arbuscular mycorrhizal fungi: A review. Science ChinaLife Sciences

532    55:474-482.

Johnson NC, Graham JH, Smith FA. 1997. Functioning of mycorrhizal associations

along the mutualism-parasitism continuum. New Phytologist 135:575-586.

Jumpponen A, Trappe JM. 1998. Dark septate endophytes: a review of facultative

biotrophic root-colonizing fungi. New Phytologist 140:295-310.

Larena I, Salazar O, Gonzalez V, Julian MC, Rubio V. 1999. Design of a primer for

ribosomal DNA internal transcribed spacer with enhanced specificity for

ascomycetes. Journal of Biotechnology 75:187-194.

Liu XY, Koba K, Makabe A, Li XD, Yoh M, Liu CQ. 2013. Ammonium first: natural

mosses prefer atmospheric ammonium but vary utilization of dissolved organic

nitrogen depending on habitat and nitrogen deposition. New Phytologist

199:407-419.

Martin DL, Bell GE, Baird JH, Taliaferro CM, Tisserat NA, Kuzmic RM, Dobson

DD, Anderson JA. 2001. Spring dead spot resistance and quality of seeded

bermudagrasses under different mowing heights. Crop Science 41:451-456.

Osono T. 2007. Ecology of ligninolytic fungi associated with leaf litter

decomposition. Ecological Research 22:955-974.

Porras-Alfaro A, Herrera J, Natvig DO, Lipinski K, Sinsabaugh RL. 2011. Diversity

and distribution of soil fungal communities in a semiarid grassland. Mycologia

103:10-21.



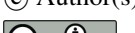

Pressel S, Bidartondo MI, Ligrone R, Duckett JG. 2010. Fungal symbioses in

        bryophytes: new insights in the twenty first century. Phytotaxa 9:238-253.

Quast C, Pruesse E, Yilmaz P, Gerken J, Schweer T, Yarza P, Peplies J, Glockner FO.

        2013. The SILVA ribosomal RNA gene database project: improved data

processing and web-based tools. Nucleic Acids Research 41:D590-D596.

        Reed SC, Coe KK, Sparks JP, Housman DC, Zelikova TJ, Belnap J. 2012. Changes to

dryland rainfall result in rapid moss mortality and altered soil fertility. Nature

        Climate Change 2:752-755.

Saikkonen K, Wali P, Helander M, Faeth SH. 2004. Evolution of endophyte-plant

        symbioses. Trends in Plant Science 9:275-280.

Schneider T, Keiblinger KM, Schmid E, Sterflinger-Gleixner K, Ellersdorfer G,

        Roschitzki B, Richter A, Eberl L, Zechmeister-Boltenstern S, Riedel K. 2012.

Who is who in litter decomposition? Metaproteomics reveals major microbial

        players and their biogeochemical functions. ISME Journal 6:1749-1762.

Simard SW, Durall DM. 2004. Mycorrhizal networks: a review of their extent,

        function, and importance. Canadian Journal of Botany 82:1140-1165.

Sperry LJ, Belnap J, Evans RD. 2006. Bromus tectorum invasion alters nitrogen

        dynamics in an undisturbed arid grassland ecosystem. Ecology 87:603-615.

Titus JH, Titus PJ, Nowak RS, Smith SD. 2002. Arbuscular mycorrhizae of Mojave

        Desert plants. Western North American Naturalist 62:327-334.



van der Heijden MGA, Horton TR. 2009. Socialism in soil? The importance of

mycorrhizal fungal networks for facilitation in natural ecosystems. Journal of

Ecology 97:1139-1150.

Walder F, Niemann H, Natarajan M, Lehmann MF, Boller T, Wiemken A. 2012.

Mycorrhizal networks: common goods of plants shared under unequal terms of

trade. Plant Physiology 159:789-797.

Wang C, et al. 2014. Aridity threshold in controlling ecosystem nitrogen cycling in

arid and semi-arid grasslands. Nature Communications (art. 4799).

Welter JR, Fisher SG, Grimm NB. 2005. Nitrogen transport and retention in an arid

land watershed: Influence of storm characteristics on terrestrial-aquatic linkages.

Biogeochemistry 76:421-440.

Weremijewicz J, Sternberg L, Janos DP. 2016. Common mycorrhizal networks

amplify competition by preferential mineral nutrient allocation to large host

plants. New Phytologist 212:461-471.

Whiteside MD, Treseder KK, Atsatt PR. 2009. The brighter side of soils: quantum

dots track organic nitrogen through fungi and plants. Ecology 90:100-108.

Whiteside MD, Garcia MO, Treseder KK. 2012. Amino Acid Uptake in Arbuscular

Mycorrhizal Plants. PLOS ONE (art. e47643).

Wu N, Zhang YM, Downing A, Aanderud ZT, Tao Y, Williams S. 2014. Rapid

adjustment of leaf angle explains how the desert moss, Syntrichia caninervis,

copes with multiple resource limitations during rehydration. Functional Plant

Biology 41:168-177.




Yahdjian L, Sala OE. 2010. Size of precipitation pulses controls nitrogen

transformation and losses in an arid Patagonian ecosystem. Ecosystems

596    13:575-585.

Yuan Y, Feng HJ, Wang LF, Li ZF, Shi YQ, Zhao LH, Feng ZL, Zhu HQ. 2017.

Potential of endophytic fungi isolated from cotton roots for biological control

against verticillium wilt disease. PLOS ONE (art. e0170557):12.

Zelikova TJ, Housman DC, Grote EE, Neher DA, Belnap J. 2012. Warming and

increased precipitation frequency on the Colorado Plateau: implications for

biological soil crusts and soil processes. Plant and Soil 355:265-282.




**Figure 1**

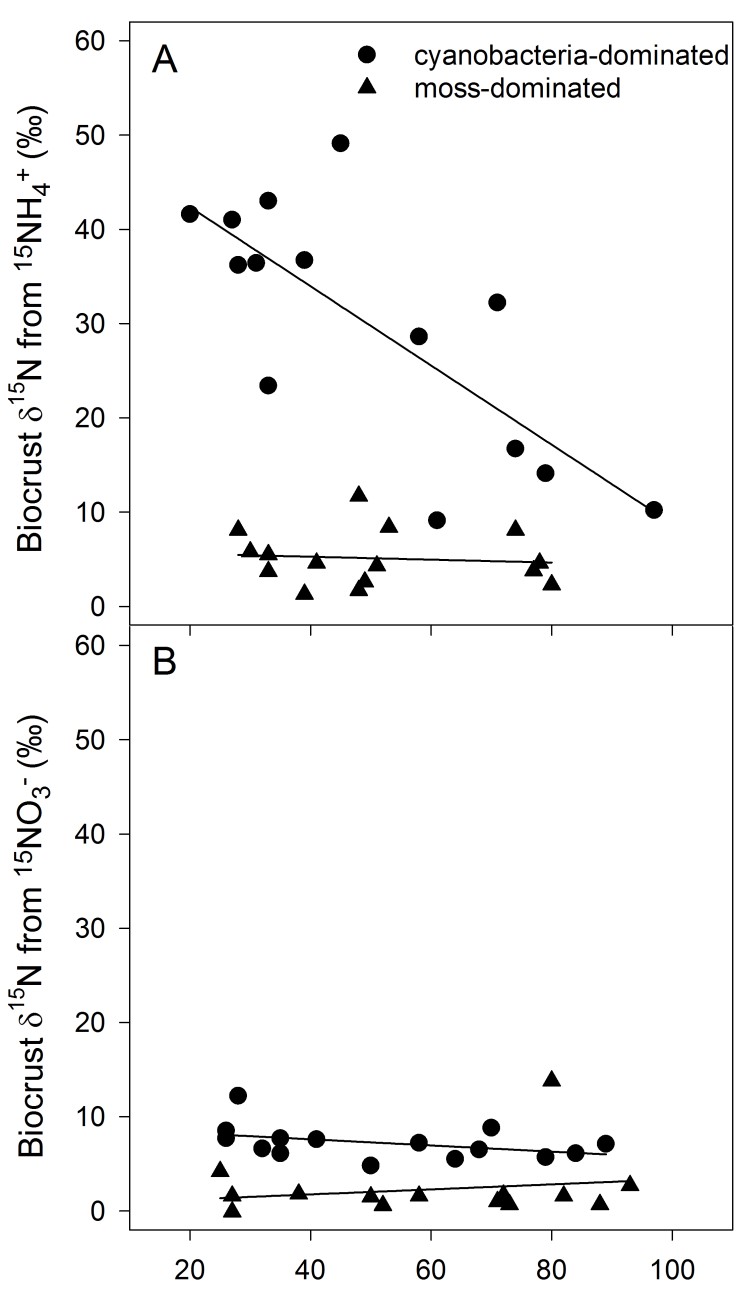



**Figure 2**

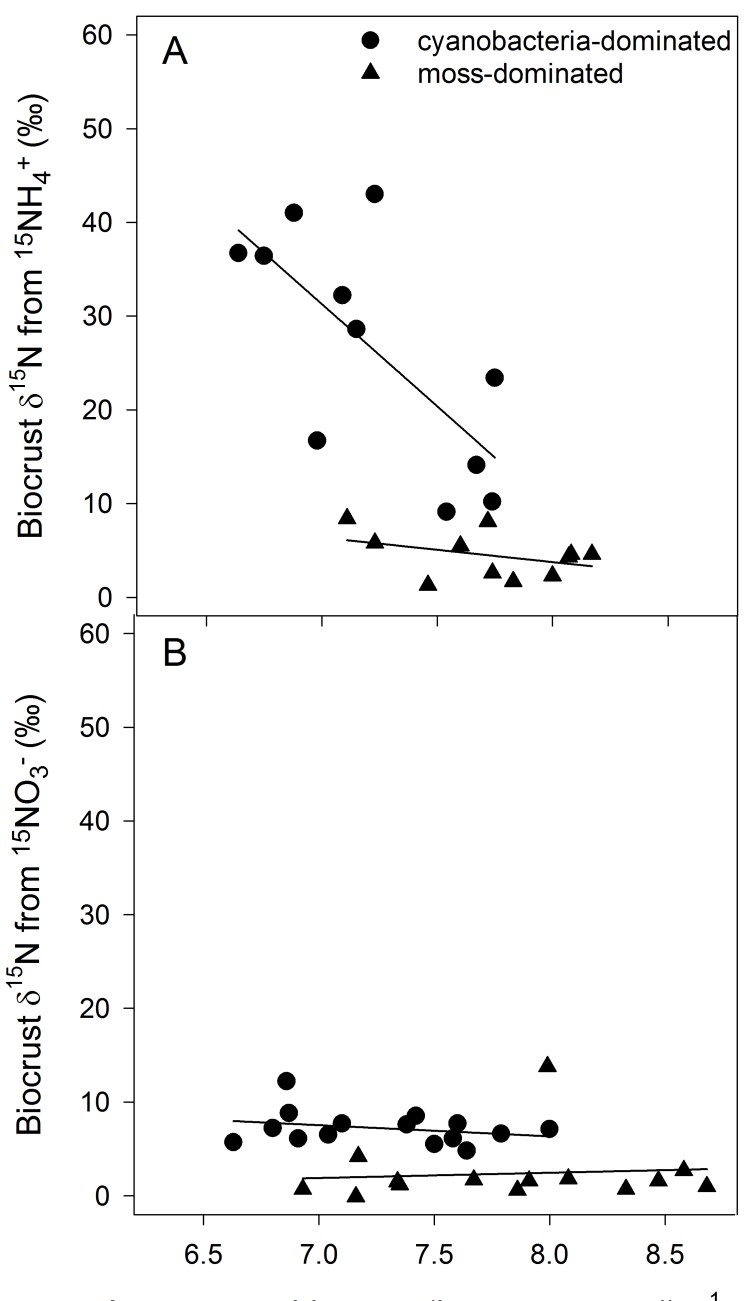

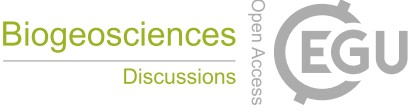



**Figure 3**

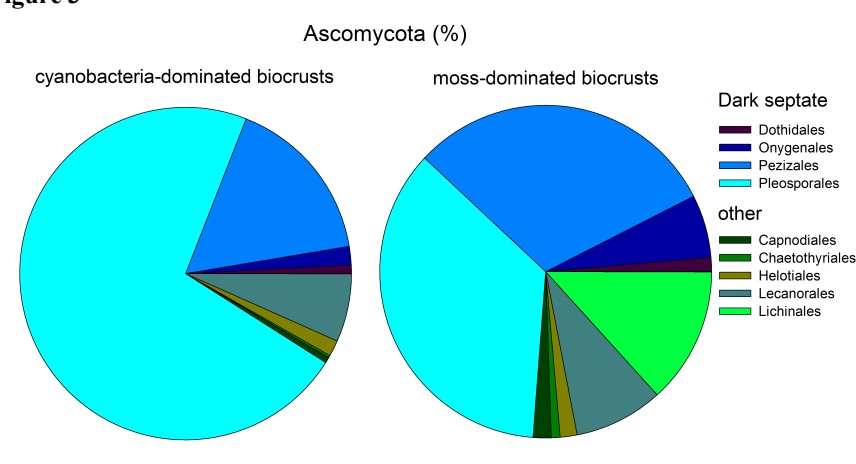
