# Peer review of "Fungal loop transfer of N depends on biocrust constituents and N form"

_Biogeosciences, 2017_

## Referee Comment (RC1) · Trent northen (Referee) · 10 Jan 2018

Aanderud et al simulated rainfall events including 15N to test the nitrogen portion of the fungal loop hypotheses in biological soil crusts. Focusing on the flow of nitrogen from biocrusts as mediated by fungi. These are very important systems and it is very important to gain insights into nutrient cycling. They added either 15N ammonia or nitrate to the biocrust and measured the resulting isotope distribution in the surrounding biocrusts and grass after 24 hours. In addition, they assessed the fungal community structure and also used qPCR to assess the biomass contributions of the major fungal community members. Overall, this is a well written and interesting manuscript addressing an important hypothesis.

[Figure]

Major comments:

-It would be helpful to include figure showing the layout of the experiment. Along these lines, might be nice to include photos of plots and types of biocrust as supplemental material

-It is a bit surprising that no tracer is found in the plant, presumably because the plants didn't wake-up with the small wetting event as you suggest. Since this is an important component of the loop hypothesis it would be worth adding additional discussion esp. on what is known about the intensity of rainfall required for plants to become active.

Minor points:

- line 59: consider specifying that they are 'terrestrial' cyanobacteria since you haven't introduced biocrusts yet

- lines 88-92: Hard to follow

- line 149: What are the molar concentrations of $K^{15}NO_3$ and $(^{15}NH_4)_2SO_4$ used?

- line 169: should be isotope 'ratio' not ration

- lines 327-328: "mosses" appears twice

Was biocrust isotope enrichment analyzed at 0 cm radial distance (at the application point)? This would be a good comparison to even see how much of the label actually remained in the soil ($NH_4^+$ vs $NO_3^-$).

To further understand the role of fungi in nutrient transport, it would be interesting to see what form the N is in at the various locations from the site of application. Though I expect this would be future research.

―――――――――――――――――――――

---

## Referee Comment (RC2) · Anonymous Referee #2 · 20 Feb 2018

**Review of Aanderud et al.: Fungal loop transfer of N depends on biocrust constituents and N form**

In their research article "Fungal loop transfer of N depends on biocrust constituents and N form" Aanderud and co-authors investigate the potential presence of a fungal connection, which may facilitate the transport of nutrients and carbohydrates between dryland vegetation constituents. This is a highly interesting research question and there have been several publications on this research hypothesis during the last years. While in some studies the rapid movement of $^{15}NO_3^-$ in a root-free environment could be shown, other studies are by far less clear and demonstrate the necessity of further research activities.

In the current study, $^{15}NH_4^+$ and $^{15}NO_3^-$ were applied to two different types of biocrusts and the occurrence of $^{15}N$ in the biocrust and grasses at varying distances away from this spot were subsequently analyzed. Unfortunately, there are some major drawbacks in this approach.

1. There are no controls, where unlabeled $NH_4^+$ and $NO_3^-$ were applied and analyzed.
2. The number of true replicates per crust type is only 3 and thus rather small. The authors calculate with an n of 14 and 15 for cyanobacteria- and moss-dominated biocrusts; however, most of the samples are not independent (since related to the same source, where $^{15}N$ was applied). Thus, calculations have to be conducted accordingly, distinguishing between pseudo- and true replicates!
3. A 2.5 mm rainfall equivalent was only applied to the central 5 cm diameter circle, whereas the area outside remained dry. As grasses and crusts at a distance of 29-120 cm and 22-97 cm, respectively, were collected, the amount of water may have been not sufficient to activate such a large area around the central biocrust. The result, that a transport was observed in cyanobacteria- but not in moss-dominated biocrusts, supports this assumption, as moss-dominated biocrusts are known to need by far more water to be activated than cyanobacteria-dominated biocrusts.

Other major comments:

1. The authors show that in cyanobacteria crusts, the gene copy number of Ascomycota decreases parallel to the $\delta^{15}N$ from $^{15}NO_3^-$. However, this does not proof that there is a functional relation between both parameters. In moss-dominated biocrusts, gene copy numbers of Ascomycota are similarly high, but no relationship to $\delta^{15}N$ was found. In the discussion (line 303ff.) it is suggested, that fungi, likely dark septate endophytes, rapidly translocate N at a rate of 40 mm h-1. I did not find any proof for this assumption and thus it needs to be phrased VERY carefully, showing that this is mainly speculation. There are fungi within the biocrust, but a proof for their functional role was not given here. How can one be sure that fungi and not e.g. bacteria or just diffusion are relevant for the transport of nutrients in this experiment?
2. In line 333 f. it is stated that "mosses may be effective scavengers for N and outcompete fungal endophytes for newly fixed N". This indeed is very speculative, as there have been several publications on mosses, showing that N compounds are strongly leached during

major rainfall events (Coxson, 1991; Coxson et al., 1992). In addition, mosses are frequently associated with cyanobacteria, which also fix and deliver nitrogen (Rousk et al., 2013, 2017). Thus, it is hard to believe that the mosses actively scavenge and hold the N compounds. I consider it as much more likely, that the amounts of water/liquid were not enough to activate the moss-dominated biocrusts at a large enough diameter.

3. Line 350 f.: A dominance of dark septate fungi does not allow any conclusion on their role in transport processes. This needs to be shown in an experimental approach or phrased much more carefully.

4. Line 376: The theory that Pleosporales are the most likely conduits is based on theory and speculation, as their functional role has not been tested. This needs to be made clear.

5. In the manuscript, gene copy numbers are seen equivalent to biomass. This is not really correct. By means of qPCR one can get an idea regarding the relevance of the different organism groups but this information is by no means equal to biomass. For example, qPCR also does not distinguish between genetic material of living and dead organisms. Thus, it is much more appropriate to speak of gene copy numbers.

As there are several major drawbacks in this experimental approach, the manuscript cannot be published in its present form. The experiments need to be analyzed in an adequate form (considering 3 replicates per crust type). It also is necessary to reassure that the different results obtained for different biocrusts are caused by functional differences and not by water limitation. The presence of organisms (determined by qPCR) does not allow conclusions on their functional roles.

Minor comments:

Line 64: *Bouteloua* sp. (instead of *Bouteloua* species)

Line 80 ff.: "In such loops, minor rainfall events may stimulate N2 fixation by free or lichen-associated cyanobacteria (Belnap et al. 2003), N mineralization by bacteria and fungi (Cable and Huxman 2004, Yahdjian and Sala 2010) and nitrification and possibly denitrification (Wang et al. 2014) all increasing the levels of $NH_4^+$ or $NO_3^-$": I think one cannot say exclusively that levels of $NH_4^+$ and $NO_3^-$ will increase during all of these processes. During nitrification for example, $NO_3^-$ is expected to decrease, but $NH_4^+$ will decrease and during denitrification $NO_3^-$ amounts are expected to decrease. Thus, I think one can say that that $NH_4^+$ and $NO_3^-$ levels will be affected by these processes, but the overall direction of change depends on the relevance of the different processes involved.

Line 329 ff.: The authors state that "When *S. caninervis* was lost from this system, a dramatic increase in $NH_4^+$, which ultimately nitrifies to $NO_3^-$, was observed". I had a look in the publication of Reed et al. (2012), and there a decrease of $NH_4^+$ and an increase of $NO_3^-$ was reported.

Line 336: There are no "stem cells" in bryophytes as they are thallophytes and not cormophytes!

Literature cited:

Coxson DS (1991) Nutrient Release from Epiphytic Bryophytes in Tropical Montane Rain-Forest (Guadeloupe). Canadian Journal of Botany-Revue Canadienne De Botanique 69: 2122-2129.

Coxson DS, McIntyre DD, Vogel HJ (1992) Pulse release of sugars and polyols from canopy bryophytes in tropical mountain rain forest (Guadeloupe, French West Indies). Biotropica 24(2a): 121-133.

Rousk K, Jones DL, DeLuca TH (2013) Moss-cyanobacteria associations as biogenic sources of nitrogen in boreal forest ecosystems. Frontiers in Microbiology. Doi: 10.3389/fmicb.2013.00150.

Rousk K, Laerkedal Sorensen P, Michelsen A (2017) Nitrogen fixation in the High Arctic: a source of 'new' nitrogen? Biogeochemistry 136: 213-222.

---

## Author Comment (AC1) · 14 Mar 2018

Authors response-general comment: Thank you for the insights and helping to improve the manuscripts. Hopefully we have addressed all your concerns adequately.

1. It would be helpful to include figure showing the layout of the experiment. Along these lines, might be nice to include photos of plots and types of biocrust as supplemental material

Authors response: We now include a supplemental figure detailing the study design with photos of both crust-types. See attached supplemental figure 1.

2. It is a bit surprising that no tracer is found in the plant, presumably because the plants didn't wake-up with the small wetting event as you suggest. Since this is an

important component of the loop hypothesis it would be worth adding additional discussion esp. on what is known about the intensity of rainfall required for plants to become active.

Authors response: We now include the following paragraph in the discussion. Good idea, thanks. 4.4 No N translocation to grass Due to the discrete nature of our minor, localized rainfall event, we were not surprised that none of the label entered the leaves of A. hymenoides. In the fungal loop hypothesis, a larger rainfall event triggers the plant to become a sink for the N building up in fungi over previous minor rainfall events. We conducted our experiment absent of a larger rainfall event and our 2.5 mm rainfall event was applied over a 5 cm diameter circle of soil in early summer. When a similar precipitation event size (2 mm) was applied across a much larger area (4 x 4 m2 plot) on Colorado Plateau soils during spring or summer, the predawn water potential of A. hymenoides was similar one day after and one day prior to watering (Schwinning et al. 2003). Thus, our minor rainfall event most likely failed to alter the water status of the grass or cause the grass to become a sink for N. Alternatively, the isotopic signal potentially became too depleted as it traveled through biocrust constituents or was adsorbed by soils to sufficiently be acquired by A. hymenoides roots and translocated to leaves. In April, at the time of the experiment, A. hymenoides was photosynthetically active. If we had added more label or evaluated the isotope signature of roots, we may have detected the 15N label in the grass tissue.

Minor points line 59: consider specifying that they are 'terrestrial' cyanobacteria since you haven't introduced biocrusts yet

Authors response: done

lines 88-92: Hard to follow

Authors response: We have edited the sentence by including a couple guide words such as host plant and rather than, and added a comma. Hopefully this helps. The sentence now reads, "Larger rainfall events may then activate plants, allowing the host

plant to receive N from the fungi, and transfer photosynthate to the fungal endophyte. If fungal endophytes are poor competitors for newly released N, preferentially sequester one inorganic N form over another, or more efficiently transform and transport $NH_4+$ rather than $NO_3-$, biocrust constituents and N form may influence the translocation of N in fungal loops."

line 149: What are the molar concentrations of $K15NO_3$ and $(15NH_4)_2SO_4$ used?

Authors response: We now include the following sentence in the methods, "For the isotopic applications, either 2.60 g of 99 at.% $K15NO_3$ or 1.70 g of 99 at.% $(15NH_4)_2SO_4$ was dissolved in 18 mL of deionized water to create a 1.43 M or 0.72 M solution respectively."

line 169: should be isotope 'ratio' not ration

Authors response: done

lines 327-328: "mosses" appears twice

Authors response: thanks

Was biocrust isotope enrichment analyzed at 0 cm radial distance (at the application point)? This would be a good comparison to even see how much of the label actually remained in the soil ($NH_4+$ vs $NO_3-$).

Authors response: We agree, but were concerned that we might fail to recapture all of the 15N in the localized rainfall applications after 24 hours. The 2 cm diameter rainfall simulations had uneven edges due to the biocrust topography. If we missed some of the label inadvertently we would falsely conclude that more 15N moved than it did. We decided against it.

To further understand the role of fungi in nutrient transport, it would be interesting to see what form the N is in at the various locations from the site of application. Though I expect this would be future research. Authors response: We agree. To

highlight your point and help others see a wonderful future endeaver we added the following sentence in the discussion right before the closing sentience of section 4.2. The statement now reads, "Unfortunately, based on our design, we were unable to distinguish the form of N captured or translocated by biocrust constituents. More information is needed to identify the importance of N form and the movement of organic N within fungal loops."

Please also note the supplement to this comment:
https://www.biogeosciences-discuss.net/bg-2017-433/bg-2017-433-AC1-supplement.pdf
* * *
[Figure]

**Supplemental Figure 1** The field design of circular plots (radius = 1.0 m) receiving minor, localized rainfall events. An example of five biocrust (BC) and *Achnatherum hymenoides* (*AH*) samples occurring randomly along eight axes (e.g., N, NE, E, SE, S, SW, W, and NW) radiating from the center of each plot (A). An isotopically labeled (0.30 mg $^{15}$N) rainfall event (2.5 mm) was sprayed onto a central circle (radius = 5 cm). Rainfall events occurred on the surfaces of cyanobacteria-dominated (B) and moss-dominated biocrust.

[Figure]

**Fig. 1.**

---

## Author Comment (AC2) · 14 Mar 2018

Authors response-general comment: First off, we strongly believe as authors that the major concerns outlined by reviewer 2 are not fatal flaws. . Hopefully we have addressed all your concerns adequately.

1. 1. There are no controls, where unlabeled NH + and NO - were applied and analyzed.

Authors response: Why is this a concern? Yes, there are some thermodynamic and enzymatic complications associated with isotopic additions, as heavier isotopes do fractionate, but we are just using the 15N as a tracer. We always refer to increases of 15N as 15N from 15NH4+, not 15NH4+. Further to help clarify any concerns over our

isotope use we added the following sentence in the discussion, Unfortunately, based on our design, we were unable to distinguish the form of N captured or translocated by biocrust constituents. More information is needed to identify the importance of N form and the movement of organic N within fungal loops." We are using enriched 15N as a tracer as many of the authors have done multiple times in the past. We are happy to provide a list of our papers using 15N as a tracer without unlabeled controls. Again, we are unclear why a control is needed. We are happy to discuss this point further.

1. 2. The number of true replicates per crust type is only 3 and thus rather small. The authors calculate with an n of 14 and 15 for cyanobacteria- and moss-dominated biocrusts; however, most of the samples are not independent (since related to the same source, where 15N was applied). Thus, calculations have to be conducted accordingly, distinguishing between pseudo - and true replicates.

Authors response: A precedent for the design was established by Green et al. 2008 in the Journal of Ecology to capture movement of tracers in biocrusts (see citation below). Our design/linear regression stats mirror their methods. I am not sure how we could even reduce our data to n=3 since all the biocrust soil and naturally occurring A. hymenoides samples are randomly distributed across many different distances. We are primarily interested in distance the isotope may travel. We decided that labeling a 5 cm diameter soil circle, via a rainfall event, and randomly selecting soils and plants up to almost 2 m away would more than serve as an indicator of movement. The potential area for the isotope to move was 1600-times (area of the entire circular plot/area of the rainfall addition) that of the area with the label. We are excited that we captured the translocation of the label and are confident in our data and analyses.

Green LE, Porras-Alfaro A, Sinsabaugh RL. 2008. Translocation of nitrogen and carbon integrates biotic crust and grass production in desert grassland. Journal of Ecology 96:1076-1085.

1. 3. A 2.5 mm rainfall equivalent was only applied to the central 5 cm diameter circle,

whereas the area outside remained dry. As grasses and crusts at a distance of 29-120 cm and 22-97 cm, respectively, were collected, the amount of water may have been not sufficient to activate such a large area around the central biocrust. The result, that a transport was observed in cyanobacteria- but not in moss-dominated biocrusts, supports this assumption, as moss- dominated biocrusts are known to need by far more water to be activated than cyanobacteria- dominated biocrusts. The experiments need to be analyzed in an adequate form (considering 3 replicates per crust type). It also is necessary to reassure that the different results obtained for different biocrusts are caused by functional differences and not by water limitation.

Authors response: Our intention was never to activate the entire biocrust and outside the 5 cm diameter circle of soil. We only activated the biocrust within the minor, localized rainfall event. We are testing a portion of the fungal-loop hypothesis. To help clarify this misconception, we added the word "dry" to the abstract and first sentence of the last paragraph in the introduction, "Minor rainfall events may allow fungi to act as conduits and reservoirs for N. To investigate the potential for biocrust constituents and N form to influence the movement of N through the putative fungal loops, we created minor, localized rainfall events and measured 15N, from 15N-NH4+ and 15N-NO3-, within the surrounding dry cyanobacteria- and moss-dominated crusts, and grass, Achnatherum hymenoides (Indian ricegrass)." Further, the rainfall event was adequate in activating the cyanobacteria and mosses in that 5 cm diameter and to stress that the rainfall event was localized we added the words "minor, localized rainfall event," which was stated in the introduction to be included in the methods, results, and discussion sections. We also state in the discussion that Syntrichia caninervis is stimulated by minor rainfall events and cite papers published by us. Here is what we now say in the discussion, "Mosses, including S. caninervis, are stimulated by minor rainfall events (Wu et al. 2014), with events as small as 1 mm activating moss photosynthesis (Coe et al. 2012). Our moss, S. caninervis, became photosynthetically active following the 2 mm rainfall event, changing in color from brown to green. Thus, S. caninervis may have absorbed the N applied in our simulated rainfall event preventing most of the isotopic

label from reaching other biocrust constituents."

Wu N, Zhang YM, Downing A, Aanderud ZT, Tao Y, Williams S. 2014. Rapid adjustment of leaf angle explains how the desert moss, Syntrichia caninervis, copes with multiple resource limitations during rehydration. Functional Plant Biology 41:168-177.

Coe KK, Belnap J, Sparks JP (2012) Precipitation-driven carbon balance controls survivorship of desert biocrust mosses. Ecology 93: 1626-1636

We have also integrated another reasoning for the lack of movement in moss-dominated crusts to help clarify our message. Line 342 states, "Additionally, moss-dominated biocrusts contain far more biomass than cyanobacteria-dominated crusts. The higher levels of biomass alone may have retained the label contributing to the lack of N movement." Again, we expected water limitations to occur outside the minor rainfall even and were testing for movement of N out of the localized rainfall event.

The presence of organisms (determined by qPCR) does not allow conclusions on their functional roles.

Authors response, please see the explanation for this concern in the next paragraph

Other major comments 2. 1. The authors show that in cyanobacteria crusts, the gene copy number of Ascomycota decreases parallel to the $\delta15N$ from $15NO3$. However, this does not proof that there is a functional relation between both parameters. In moss-dominated biocrusts, gene copy numbers of Ascomycota are similarly high, but no relationship to $\delta15N$ was found. In the discussion (line 303ff.) it is suggested, that fungi, likely dark septate endophytes, rapidly translocate N at a rate of 40 mm h-1. I did not find any proof for this assumption and thus it needs to be phrased VERY carefully, showing that this is mainly speculation. There are fungi within the biocrust, but a proof for their functional role was not given here. How can one be sure that fungi and not e.g. bacteria or just diffusion are relevant for the transport of nutrients in this experiment?

Authors response: We agree that an estimation of biomass does not indicate function.

[Figure]

In the introduction, last paragraph, we state our intentions, "In tandem with 15N analyses, we estimated the biomass of two major division of fungi (Ascomycota and Basidiomycota) and bacteria, and characterized fungal communities by sequencing the 18S rRNA gene to identify potential links between fungal taxa and 15N movement." To address your concern and be more explicit in the text we made the following edits: line 263, 272 in the results, added proxy or estimate when referring to biomass; line 307 in the discussion, added "may stimulate" to "rainfall event stimulated"; and added the following sentence to discussion section line 376, "However, to verify the movement of N through Ascomycota and the role of biomass in translocation, a more direct approach is needed. For example, quantum dots (fluorescent nanoscale semiconductors) have tracked the flow of organically derived N into arbuscular mycorrhizae and into Poa annua in less than 24 hours (Whiteside et al. 2009)."

2. 3Line 350 f.: A dominance of dark septate fungi does not allow any conclusion on their role in transport processes. This needs to be shown in an experimental approach or phrased much more carefully.
 2. 4. Line 376: The theory that Pleosporales are the most likely conduits is based on theory and speculation, as their functional role has not been tested. This needs to be made clear.

Author response: We were sensitive to any concrete statement about Pleosporales role. For example, in the discussion the first sentence of the paragraph on Pleosporales states, "Our results support the idea that Pleosporales are the most likely conduits for N. However, to address your concern we have have revised/rephrased the statements concerning Pleosporales involvement throughout the manuscript. We understand that it is a theory but it is not speculation but a rational conclusion based on one of the first high-throughput sequencing efforts of eukaryotes in biocrust soils. Specifically, we altered the concluding sentence in the abstract to now read, "Our findings suggest that minor rainfall events may allow dark septate Pleosporales to rapidly translocate N in the absence of a plant sink." Instead of "allowed;" and we added the following statement in the discussion section on Pleosporales that direct states Pleosporales

participation in fungal loops as a theory, "Further research is needed to address the theory of Pleosporales conduits in biocrusts and the ecological importance of the dark septate endophytes in desert systems."

2. 2. In line 333 f. it is stated that "mosses may be effective scavengers for N and outcompete fungal endophytes for newly fixed N". This indeed is very speculative, as there have been several publications on mosses, showing that N compounds are strongly leached during major rainfall events (Coxson, 1991; Coxson et al., 1992).

Coxson DS (1991) Nutrient Release from Epiphytic Bryophytes in Tropical Montane Rain -Forest (Guadeloupe).CanadianJournalofBotany-RevueCanadienneDeBotanique69:2122-2129.

Coxson DS, McIntyre DD, Vogel HJ (1992) Pulse release of sugars and polyols from canopy bryophytes in tropical mountain rain forest (Guadeloupe, French West Indies). Biotropica 24(2a): 121-133.

Authors response: Thank you for the references and the insight. We have altered the text substantially to better convey/explain our intent. We no longer refer to mosses competing with fungi for N. We now state, "The ability of mosses to scavenge atmospheric deposited N is well recognized in other systems (Liu et al. 2013, Fritz et al. 2014). Most mosses acquire N from either wet and dry atmospheric deposition (Yaunming et al. 2016) or as biogenic sources from cyanobacterial associations on their leave (Rousk et al. 2013). Mosses, including S. caninervis, are stimulated by minor rainfall events (Wu et al. 2014), with events as small as 1 mm activating moss photosynthesis (Coe et al. 2012). Our moss, S. caninervis, became photosynthetically active following the 2 mm rainfall event, changing in color from brown to green. Thus, S. caninervis may have absorbed the N applied in our simulated rainfall event preventing most of the isotopic label from reaching other biocrust constituents." Authors response: am not sure that our minor, localized rainfall events qualifies as a major rainfall event. Further, both of the Coxson papers are on epiphytic bryophytes/canopy bryophytes in tropical

systems. I know that this goes without saying but cold deserts are extremely different from rainforest. Most importantly, the amount of N compounds moving through aerial mosses on tropical trees is very different that the N compounds in classically N-poor, arid systems. We definitely need more research on moss physiology across biomes.

In addition, mosses are frequently associated with cyanobacteria, which also fix and deliver nitrogen (Rousk et al., 2013, 2017).

Rousk K, Jones DL, DeLuca TH (2013) Moss-cyanobacteria associations as biogenic sources of nitrogen in boreal forest ecosystems. Frontiers in Microbiology. Doi : 10.3389/fmicb.2013.00150.

Authors response: The predominant, and really only moss at our site, Syntrichia caninervis, most likely receives N from cyanobacteria that live on its leaves and dry and wet deposition, but very little from the soils due to not possessing true roots. Thus, during our wet deposition event, rainfall event, we hypothesize that Synthricia was effectively holding N from the rainfall event in our N-limited, semi-arid system. We now include Rousk et al. 2013 in our manuscript, see sentence in the discussion "Most mosses acquire N from either wet and dry atmospheric deposition (Yaunming et al. 2016) or as biogenic sources from cyanobacterial associations on their leave (Rousk et al. 2013)."

Thus, it is hard to believe that the mosses actively scavenge and hold the N compounds. I consider it as much more likely, that the amounts of water/liquid were not enough to activate the moss-dominated biocrusts at a large enough diameter.

Author response: We definitely activated the biocrusts/mosses in our 5 cm diameter rainfall event. Previous studies show 1mm is sufficient (Coe et al. 2012 ); in addition, they turned green with our water addition. For more explanation, see above our response to 1.3

5. In the manuscript, gene copy numbers are seen equivalent to biomass. This is not really correct. By means of qPCR one can get an idea regarding the relevance of the

different organism groups but this information is by no means equal to biomass. For example, qPCR also does not distinguish between genetic material of living and dead organisms. Thus, it is much more appropriate to speak of gene copy numbers.

Authors response: we never said that it was equivalent to biomass and talk of it as a proxy or estimate of biomass. We are well aware of the potential artifacts associated with PCR but our division specific primers to estimate specific fungi in complex matrices is a wonderful technique and gene copy number of an organism is a common proxy for biomass in microbial ecology. But to more explicitly clarify any misconceptions we made the following edits: line 263, 272, 358, and 364 added proxy or estimate when referring to biomass. For exampleline 359 states, "We found that NH4+, but not NO3-, was rapidly translocated within crusts. The enrichment of 15N, from 15NH4+, in cyanobacteria biocrusts was related to our proxy for Ascomycota biomass and potentially dark septate fungi due to their dominance in our sequencing effort." We are happy to provide references of gene copy number being used as a proxy for biomass.

Minor comments:
 Line 64: Bouteloua sp. (instead of Bouteloua species)

Authors response: Thanks

Line 80 ff.: "In such loops, minor rainfall events may stimulate N2 fixation by free or lichen -associated cyanobacteria (Belnap et al.2003), N mineralization by bacteria and fungi (CableandHuxman2004, Yahdjian and Sala 2010) and nitrification and possibly denitrification (Wang et al. 2014) all increasing the levels of NH4 or NO3: I think one cannot say exclusively that levels of NH4 and NO3 will increase during all of these processes. During nitrification for example, NO3 is expected to decrease, but NH4 will decrease and during denitrification NO3 amounts are expected to decrease. Thus, I think one can say that that NH4 and NO3 levels will be affected by these processes, but the overall direction of change depends on the relevance of the different processes involved.

Authors response: we have altered the sentence to the following, In such loops, minor

rainfall events may stimulate N2 fixation by free or lichen-associated cyanobacteria (Belnap et al. 2003), N mineralization by bacteria and fungi (Cable and Huxman 2004, Yahdjian and Sala 2010), and nitrification and possibly denitrification (Wang et al. 2014) potentially altering levels of NH4+ or NO3-." Thanks for the catch

Line 329 ff.: The authors state that "When S. caninervis was lost from this system, a dramatic increase in NH4, which ultimately nitrifies to NO3, was observed". I had a look in the publication of Reed et al. (2012), and there a decrease of NH and an increase of NO was reported.

Authors response: Thanks, I can see how what we stated was a little confusing but I believe we are saying the same thing. One of our authors was also an author on this paper. We have now changed the text to read, "When S. caninervis was lost from our system, a dramatic increase in nitrification rates were observed (Reed et al. 2012). The higher levels of nitrification were most likely supported by the decomposition of dead moss biomass and subsequent release of new NH4+; however, after the moss mortality, inorganic N pooled as NO3-, in the remaining cyanobacteria-dominated biocrust." We believe the edits address your concnern.

43
Line 336: There are no "stem cells" in bryophytes as they are thallophytes and not cormophytes.

Authors response: Thanks, we removed "stem cells" from the sentence

---

## Author Response (AR1)

Editor Comment 1: Reviewer 2 notes that statistical analyses should be improved by considering that most measurements are related to the same source, where 15N was applied. Thus, calculations have to be conducted accordingly, probably by including distance, biocrust type and their interaction in the analysis, as well as experimental plot as random factor or similar. This should be feasible given your experiment design

We have substantially altered the statistical models in the manuscript. We now use mix effects linear models to assess the relationships between 15N signatures in biocrusts and grass leaves relating distance and our proxy for Ascomycota biomass. As suggested by the editor, we now treat the experimental plot as a random effect in all models and directly addresses reviewers 2 main concern accounting for the variation due to sampling biocrusts of grasses within three circular plots. We now nest our distances and Ascomycota gene copy number within each experimental plot. The models are also now included in the manuscript for the reader to see the relationships of all variables involved. The inclusion of the model decreased the significance of our findings but did not substantially alter any of our major findings-NH4+ moved in cyanobacteria-dominated biocrusts and Ascomycota most likely involved. Our initial finding of 15NO3 movement in cyanobacteria was dropped and we have altered the results section to reflect all changes.

We feel that our model changes most accurately present our data and appropriately treat our distance variable. Only one crust type and one N form lead to positive results. There is no benefit to creating an overall model with N form and crust type. If needed we are willing to add another model if the editor still believes it is necessary.

Editor comment 2: Low amount of water added (only 2mm) has been also presented as an important factor that may affect results obtained in moss dominated biocrust. Following reviewer two suggestions authors must, at least, recognize this issue in the manuscript and also moderate related statements, as well as other speculations within the manuscript that are not proven by the experimental data, some of them already explained in the discussion.

We have substantially altered the discussion section to address the low amount of our rainfall event and moss biocrust activity. We now include the following new paragraph.

"The lack of $^{15}$N movement in moss-dominated crusts may reside in the nature of our minor rainfall event. Our moss, *S. caninervis*, became photosynthetically active following the 2 mm rainfall event, changing in color from brown to green, but only in the discrete biocrust patches that we watered. Mosses, including *S. caninervis*, are stimulated by minor rainfall events (Wu et al. 2014), with events as small as 1 mm activating moss photosynthesis (Coe et al. 2012). Our rainfall event was intended to wet a small circle of biocrust to a depth of 1 cm. However, the additional aboveground biomass of mosses and the rugose topography of moss-dominated crusts relative to the smooth cyanobacteria-dominated crusts may have limited the depth our minor rainfall event penetrated the soil and, in turn, activated other biocrust components. Also, water from our event might have evaporated more quickly from the mossy biocrust surface, limiting the activity time of all constituents involved. To more conclusively determine the

potential for fungal loops to exist in moss-dominated biocrusts, more information is needed to determine the importance of effective rainfall size in initiating fungal loops."

We have also tempered our statements further in the text by incorporating statements like "the seeming lack of loops in moss-dominated crusts may stem…" We believe that our edits temper our previous message and definitely highlight the concerns surrounding the limitations of the inferences we can make from our design.

---

## Author Response (AR3)

We have included all of the minor comments and thank you.

Here is our response to Referee #2

Referee #2
There is one issue, which is not clear. Whereas in Fig. 1a 14 cyanocrusts were analyzed regarding distance and delta 15N, in Fig. 2a only 11 cyanocrusts were plotted. I wonder what has happened to the remaining samples.

Authors' response: Some of the DNA extracted for both biocrust types were tough to amplify. For clarity we now include the following text in the sentence starting line 211):

"…combination (2 biocrust types × 3 circular plots locations × 2 N forms × ≈ 1 – 5 biocrusts = 48). DNA from eight moss-dominated and four cyano-dominated biocrusts were difficult to amplify, especially one moss biocrust circular plot reducing the number of replicants from five to one."

We believe this appropriately resolves the issue and thanks for the help.